# Prognostic Factors and Talaporfin Sodium Concentration in Photodynamic Therapy for Recurrent Grade 4 Glioma

**DOI:** 10.3390/ph18040583

**Published:** 2025-04-16

**Authors:** Mikoto Onodera, Shuji Kitahara, Yasuto Sato, Takakazu Kawamata, Yoshihiro Muragaki, Ken Masamune

**Affiliations:** 1Faculty of Advanced Techno-Surgery (FATS), Institute of Advanced Biomedical Engineering and Science, Tokyo Women’s Medical University, 8-1 Kawada-Cho, Shinjuku, Tokyo 162-8666, Japan; onodera.mikoto@twmu.ac.jp (M.O.); muragaki@people.kobe-u.ac.jp (Y.M.); masamune.ken@twmu.ac.jp (K.M.); 2Department of Pharmacy, Tokyo Women’s Medical University, 8-1 Kawada-Cho, Shinjuku, Tokyo 162-8666, Japan; 3Department of Hygiene and Public Health, Tokyo Women’s Medical University, 8-1 Kawada-Cho, Shinjuku, Tokyo 162-8666, Japan; sato.yasuto@twmu.ac.jp; 4Graduate School of Public Health, Shizuoka Graduate University of Public Health, 4-27-2 kita Ando, Aoi-ku, Shizuoka, Shizuoka 420-0881, Japan; 5Department of Neurosurgery, Tokyo Women’s Medical University, 8-1 Kawada-Cho, Shinjuku, Tokyo 162-8666, Japan; tkawamata@twmu.ac.jp; 6Center for Advanced Medical Engineering Research and Development, Kobe University, 1-5-1 Minatojima Minamimachi, Chuo-Ku, Kobe, Hyogo 650-0047, Japan

**Keywords:** Karnofsky Performance Status, precision medicine, treatment outcome, sequence analysis, RNA, photochemotherapy, glioma, gene expression, genotype

## Abstract

**Background:** Although extensive resection improves the prognosis of gliomas, it risks impairing critical brain functions. Photodynamic therapy (PDT) utilizing talaporfin sodium (TS) targets tumor cells upon light activation. Despite its approval in Japan, TS application remains restricted, and factors influencing its efficacy are unclear. We aimed to identify TS efficacy determinants to optimize treatment outcomes. **Methods:** Data from 171 patients with grade 4 glioma who underwent surgery and PDT at Tokyo Women’s Medical University Hospital between January 2017 and March 2024 were retrospectively analyzed. Clinical variables evaluated included age, sex, genotype, Karnofsky Performance Status (KPS), serum albumin (Alb) levels, MIB-1 expression levels, and medication history. TS concentrations in tumor tissues were quantitatively assessed in 82 patients (41 primary, 41 recurrent). Survival outcomes were analyzed. RNA-seq was performed on the three highest and three lowest TS concentration samples with significant TS concentration variations to investigate corresponding gene expression changes. **Results:** Multivariate analysis identified KPS (hazard ratio [95% confidence interval]: 0.96 [0.93–0.99], *p* = 0.01) and Alb (3.68 [1.05–13.76], *p* = 0.047) as independent prognostic factors. In recurrent cases, higher TS concentrations were significantly associated with improved survival (*p* = 0.0454). RNA-seq analysis indicated decreased expression of *ACTB* and *PDPN* genes in samples with lower TS concentrations, suggesting potential resistance mechanisms. **Conclusions:** TS concentration is a critical determinant of PDT efficacy, especially in recurrent glioma, highlighting its prognostic significance. Alb may affect treatment outcomes by mediating TS binding. RNA-seq findings imply that low TS concentrations may suppress immune and stress response-related genes, potentially diminishing PDT sensitivity.

## 1. Introduction

Brain tumors comprise approximately 1.5–2% of all malignant tumors. Gliomas, particularly grade 4 glioblastoma (GBM), represent highly invasive primary malignant neoplasms of the central nervous system. The average incidence of GBM is 3.19 cases per 100,000 individuals, with a dismal 5-year survival rate of only 5% [1]. GBM is characterized by a poor prognosis, with a median overall survival (OS) of 14.5 months [2]. In older patients, survival is often shorter, typically averaging less than 8.5 months, and the median OS for recurrent GBM ranges from 6.5 to 7.6 months [3]. Although over the past 40 years substantial funding for intracranial malignant tumors has been provided by the U.S. National Institutes of Health [4], the overall survival of patients with GBM has not significantly improved over the long term.

GBM exhibits invasive growth within adjacent normal brain tissue and tends to recur in these infiltrative regions. The presence of critical functional areas within the brain, including language and motor cortices, complicates extensive tumor resection. This characteristic distinguishes GBM from tumors in other organs. The extent of tumor resection and the preservation of normal brain tissue are crucial factors significantly impacting both patient prognosis and postoperative quality of life [5]. Current treatment strategies for GBM encompass surgery, chemotherapy, and radiotherapy. Among these, photodynamic therapy (PDT) has garnered attention as a method that facilitates maximal tumor resection while preserving normal brain tissue [6,7]. PDT involves administering a photosensitizer that selectively accumulates in tumor cells, followed by irradiation with light of a specific wavelength. Under aerobic conditions, this process induces the generation of cytotoxic singlet oxygen that selectively eliminates cancerous and abnormal tissues [8].

In the 1960s, hematoporphyrin derivatives, known as Photofrins, were demonstrated to selectively accumulate in cancerous tissues [9]. In 1980, Perria et al. published a clinical report on PDT using helium/neon laser irradiation following Photofrin II administration [10], followed by numerous subsequent clinical studies [11,12]. Talaporfin sodium (TS), a second-generation photosensitizer, was first developed in Japan and received approval for early-stage lung cancer treatment in 2003 [13]. A decade later, in 2013, TS-PDT was approved for the treatment of primary malignant brain tumors by the National Health Insurance system [6,7]. Recent research indicated that surgery in conjunction with PDT significantly prolonged survival compared to that with surgical intervention alone [8].

The prognosis of GBM is influenced by four primary factors: patient age at diagnosis, tumor malignancy (genetic profile), Karnofsky Performance Status (KPS), and the extent of surgical resection [14]. However, the clinical and molecular biological factors associated with long-term survival following PDT remain unclear. Furthermore, the correlation between intratumoral drug concentration and survival requires elucidation. Previous studies have shown that the efficacy of first-generation photosensitizers on TS depends on the drug concentration; consequently, we hypothesized that the clinical efficacy of TS may similarly rely on its concentration [15]. We propose that the intraoperative rapid measurement of TS concentration could have clinical utility in predicting prognosis and evaluating tumor grade.

We hypothesized that certain factors influence the therapeutic efficacy of PDT and intratumoral TS concentration can impact patient prognosis. Accordingly, the objectives of this study were to investigate the factors associated with the efficacy of PDT and to evaluate the correlation between intratumoral TS concentration and clinical outcomes.

## 2. Results

### 2.1. Clinical Characteristics of Patients Who Underwent PDT

The clinical characteristics of the 171 patients who underwent PDT are summarized in Table 1. The cohort comprised 65.5% males and 34.5% females. Among these, 74 patients (56.8% male, 43.2% female) were newly diagnosed with the disease, while 97 (72.2% male, 27.8% female) had recurrent disease, with no significant difference in sex ratios between the groups. The mean age of the entire cohort was 51.9 years: the newly diagnosed group had an average age of 57.5 years, and the recurrent group had an average age of 47.6 years, indicating a significantly younger age in the recurrent group (*p* < 0.001).

Histopathological diagnoses of grade 4 tumors included IDH wild-type glioblastoma in 131 patients (70 newly diagnosed, 61 recurrent), IDH mutant glioblastoma in 26 patients (3 newly diagnosed, 23 recurrent), IDH mutant astrocytoma in 5 patients (0 newly diagnosed, 5 recurrent), and glioblastoma (NOS) in 9 patients.

The median KPS of the cohort was 70, with a median albumin level of 3.3 g/dL and total cholesterol (T-Cho) level of 202 mg/dL. Benzodiazepine use was reported in 10% of the patients, while 35% had a history of smoking.

With regard to tumor laterality, 41 newly diagnosed and 50 recurrent cases were located in the left hemisphere, whereas 43 newly diagnosed and 52 recurrent cases were located in the right hemisphere. Additionally, 15 patients presented with bilateral tumors (bilateral tumors were counted on both the left and right sides).

### 2.2. Identification of Risk Factors in Patients Who Underwent PDT

For both newly diagnosed (*n* = 74) and recurrent patients (*n* = 97), univariate and multivariate regression analyses were conducted to assess the impact of sex, age, KPS, albumin levels, cholesterol levels, benzodiazepine use, smoking history, and tumor location on survival.

In the newly diagnosed group, KPS and albumin levels were identified as independent and significant prognostic factors in the multivariate analysis. However, age, benzodiazepine use, and smoking history did not emerge as significant risk factors (age, hazard ratio [95% CI]: 1.02 [0.99–1.05], *p* = 0.27; benzodiazepine use, 3.06 [0.96–9.76], *p* = 0.059; and smoking history, 1.58 [0.67–3.73], *p* = 0.30) (Table 2).

In the recurrent group, analysis employing the same model indicated that only a decline in KPS represented a significant risk factor (0.96 [0.93–0.99], *p* = 0.01) (Table 2).

### 2.3. Calibration Curve for TS Concentration Measurement Using HPLC (Internal Standard Method)

A calibration curve was constructed by plotting the peak area ratio of each standard compound against the internal standard on the *Y*-axis and the concentration ratio (sample concentration/internal standard concentration) on the *X*-axis, using a straight line passing through the origin. The TS concentration was calculated based on this calibration curve.

The coefficient of determination (R^2^) was established at 0.999, indicating a high degree of accuracy and reliability of the measurement method (Figure 1). The quality of the calibration curve was characterized by the coefficient of determination (R^2^), which was established at 0.999. This value indicates a very strong linear relationship between the measured peak area ratios and the corresponding concentration ratios; however, it should be noted that R^2^ alone does not directly reflect the accuracy or reliability of the measurements. To evaluate accuracy, we assessed both the precision of the analyses (quantified by repeatability and reproducibility) and biases observed through recovery studies.

### 2.4. Impact of Intratumoral Drug Concentration on Survival Outcomes in Patients Who Underwent PDT

The median intratumoral TS concentration in 82 patients (including both newly diagnosed and recurrent cases) was 0.13226. The median overall survival (OS) was 12 months (range, 0–38 months) in the low-concentration group (below the median) and 10 months (range, 1–30 months) in the high-concentration group (above the median). The mean observation period for the entire cohort was 12.5 ± 8.8 months.

Among the 41 newly diagnosed patients, the median TS concentration was 0.16333. The median OS was 15 months (range, 4–38 months) in patients with TS concentrations below 0.16333 and 8 months (range, 1–30 months) in those with TS concentrations of 0.16333 or higher (*p* = 0.3391) (Figure 2A). The mean observation period in this group was 14.6 ± 10.2 months.

Among the 41 recurrent cases, the median TS concentration was 0.0883. The median OS was 10.5 months (range, 0–19 months) in patients with TS concentrations below 0.0883 and 9 months (range, 1–30 months) in those with TS concentrations of 0.0883 or higher (*p* = 0.0045). These results indicate that patients with TS concentrations of 0.0883 or higher had a statistically significant improvement in OS compared to those with lower concentrations (Figure 2B). The mean observation period in this group was 10.3 ± 6.6 months.

### 2.5. Analysis of Clinical Factors in the High vs. Low Intratumoral TS Concentration Groups

Among the 82 patients for whom intratumoral TS concentration was measured, the newly diagnosed group consisted of 41 patients (high concentration, 51.2%; low concentration, 48.8%) with a mean age of 60.7 years (SD: 12.2 years) and 61.3 years (SD: 11.9 years), respectively. 

Similarly, the recurrent group included 41 patients (high concentration, 51.2%; low concentration, 48.8%) with a mean age of 47.6 years (SD: 14.2 years) and 53.6 years (SD: 10.5 years), respectively. 

No significant differences in age or sex were observed between the high- and low-concentration groups (age, Newly Diagnosed *p* = 0.884 and Recurrence *p* = 0.5843, and sex, Newly Diagnosed *p* = 0.837 and Recurrence *p* = 0.5843, respectively). Additionally, no significant differences were found in tumor location, genetic mutations, benzodiazepine use, or smoking history, suggesting that no specific factors were associated with TS concentrations (Table 3).

### 2.6. RNA-Seq Analysis Identified Differential Gene Expression Associated with TS Concentration

RNA-seq was conducted to comprehensively explore the gene expression profiles associated with varying TS concentrations. Comparative differential expression analysis between the high- and low-TS-concentration groups revealed statistically significant variations in gene expression. Specifically, *FLOT1* (*p* = 0.018), *HSPA5* (*p* = 0.010), *ACTB* (*p* = 0.002), *PDIA4 (p =* 0.0007), *HMOX1* (*p* = 0.040), *PDPN* (*p* = 0.00000019), and *ROMO1* (*p* = 0.0134) were significantly downregulated in the low-TS-concentration group (*p* < 0.05), suggesting a potential correlation between TS concentration and tumor-specific gene regulation. These findings provide valuable insights into the molecular mechanisms underlying TS uptake and their potential impact on tumor biology (Figure 3).


*2.7. qPCR Analysis Identified Significant Differential Expression of ACTB*


To further validate the gene expression differences observed in the RNA-seq analysis, qPCR was performed to quantify the expression levels of *ACTB*, *CALR*, *VEGFA*, *PDPN*, *RTN4*, and *DNER*. Comparative analysis of the high- and low-TS-concentration groups revealed distinct expression patterns. The mean *ACTB* expression level was significantly higher in the high-concentration group (4.8486) than in the low-concentration group (1.4713), indicating a statistically significant differential expression. Conversely, the expression levels of *CALR* (1.0643 vs. 1.0043), *VEGFA* (1.1048 vs. 1.1239), *PDPN* (2.4902 vs. 1.3484), *RTN4* (1.0074 vs. 1.8643), and *DNER* (1.0625 vs. 1.1321) did not demonstrate significant differences between the groups. These findings suggest that *ACTB* plays a key role in TS concentration-dependent molecular mechanisms and warrant further investigation to elucidate its potential functional implications in tumor biology (Figure 4).

## 3. Discussion

### 3.1. Main Results

This study elucidated key prognostic factors associated with the efficacy of PDT in patients with grade 4 gliomas. In newly diagnosed patients, high albumin levels and low KPS were associated with prognosis. Conversely, in recurrent cases, a low KPS correlated with unfavorable outcomes. Regarding the relationship between intratumoral TS concentration and survival, a higher intratumoral TS concentration was associated with better prognosis in patients with recurrent disease, whereas no such relationship emerged in patients with newly diagnosed disease. Although no significant differences in intratumoral TS uptake were observed between newly diagnosed and recurrent cases, a tendency toward lower uptake was observed in the recurrent group. Furthermore, no specific factors significantly influenced TS uptake, indicating uniform TS uptake across all factors examined in this study.

RNA-seq analysis revealed a potential enrichment of Gene Ontology (GO) categories related to the endoplasmic reticulum, immune response, and cell membrane endocytosis among the differentially expressed genes. Although the small sample size (*n* = 6) limits the generalizability of these findings, they offer preliminary insights that may guide future investigations on the relationship between intratumoral TS uptake and immune response in gliomas, as well as the molecular mechanisms underlying the therapeutic effects of PDT. These findings also suggest that individual patient factors may play a role in modulating PDT efficacy and could contribute to refining patient selection strategies.

### 3.2. Factors Influencing the Effectiveness of TS

#### 3.2.1. Prognostic Factors

Albumin influences the half-life and tissue distribution of drugs. TS is reported to have a high affinity for albumin and LDL [16,17], and findings from this study suggest that albumin affects PDT treatment outcomes. Prior in vivo studies utilizing rat cardiomyocytes demonstrated that higher albumin concentrations reduced cell lethality [18]. This study posits that higher albumin levels may constitute a risk factor for poor prognosis following PDT. This observation implies that the strong affinity of TS for albumin could influence intracellular drug uptake, as well as its interactions with receptors and signaling cascades. Moreover, it is hypothesized that the albumin-binding ratio and the proportion of free drug may impact PDT efficacy. This hypothesis supports the notion that drug distribution within the tumor microenvironment is a key determinant of treatment efficacy [15].

In patients with glioma, survival has been reported to be associated with age, malignancy grade, extent of resection, and KPS [8,19]. This study examined patients under unified conditions regarding malignancy grade and extent of resection and revealed trends similar to those reported in previous studies regarding KPS. Although an increased risk ratio was observed with older age, consistent with previous reports, statistical significance was not attained, possibly due to limited sample size.

Prior studies have suggested that hematoporphyrin derivative (HpD), a photosensitizer, is taken up by tumor cells through LDL-mediated transport and that peripheral benzodiazepine receptors (PBRs) may facilitate drug uptake [20,21]. However, this study found that high cholesterol levels did not correlate with prognosis. Unlike HpD, TS features an aspartic acid side chain that enhances hydrophilicity and concurrently reduces affinity for LDL, suggesting a distinct uptake mechanism. Although no significant associations were evident between benzodiazepine use and prognosis, a trend emerged that warrants further investigation.

Overall, these results imply that clinical factors such as albumin levels and KPS are important in newly diagnosed cases, while KPS may represent a major prognostic factor in recurrent cases. These findings provide valuable references for establishing PDT eligibility criteria and developing treatment strategies.

#### 3.2.2. Relationship Between Intratumoral TS Concentration and Survival

The mechanism of direct PDT-induced cellular damage is posited to rely on the intracellular distribution of photosensitizers. Johansson et al. reported that among patients with GBM undergoing PDT with 5-aminolevulinic acid, those exhibiting higher intratumoral concentrations of the endogenous photosensitizer protoporphyrin IX demonstrated superior long-term outcomes [15]. Although TS represents a different photosensitizer, findings from the present study indicate a positive correlation between higher intratumoral TS concentrations and improved prognosis in recurrent cases. However, this trend was not observed in newly diagnosed patients.

In recurrent tumors, prior surgery, radiotherapy, and chemotherapy may alter the tumor microenvironment and immune response, potentially influencing drug distribution and therapeutic efficacy. The disruption of the blood–brain barrier (BBB) in recurrent tumors may enhance drug permeability, resulting in increased intratumoral drug accumulation and improved treatment outcomes for some patients [22]. In contrast, the largely intact BBB in newly diagnosed tumors may restrict drug penetration, obscuring any potential correlation between TS concentration and prognosis. Additionally, changes in blood flow dynamics due to fibrosis in recurrent tumors may affect drug distribution. Moreover, tumor biology and drug sensitivity may be altered as a result of prior treatments [23].

Another factor that may influence intratumoral TS levels is the expression of drug efflux transporters, such as ABCG2, which actively exports porphyrin-based photosensitizers from tumor cells, thereby lowering intracellular drug concentrations. Variability in ABCG2 expression between tumors or in response to previous therapies may further contribute to heterogeneous TS distribution and potentially influence PDT efficacy. The recurrent patient cohort comprised individuals who survived initial treatment, potentially including a subgroup demonstrating heightened therapy responsiveness. In contrast, in newly diagnosed patients, tumor progression and treatment responsiveness may exert a greater influence on prognosis, rendering the correlation between drug concentration and outcome statistically less apparent. The limited sample size may also have contributed to the challenges in detecting this association.

Cumulatively, these findings suggest a pronounced association between drug accumulation and prognosis in patients with recurrent disease, whereas tumor characteristics and treatment effects may have obscured this relationship in patients with newly diagnosed disease. Although no clinical systems currently exist to measure intratumoral TS concentration in real time, our findings indicate the potential clinical value of intraoperative measurement. Given that TS concentration results can be obtained within approximately 1 h, it may become feasible to estimate prognosis by the end of surgery in recurrent cases.

This analysis suggests that TS concentration assumes greater significance in recurrent cases. These results imply that intracellular drug concentration, alongside KPS, a recognized prognostic factor for recurrent GBM, may serve as a prognostic predictor.

#### 3.2.3. Drug Uptake and Intracellular Dynamics in PDT

The uptake, intracellular distribution, and retention of porphyrin-based photosensitizers in tumor cells are influenced by a variety of cellular and microenvironmental factors. These include the proliferative state of the cells, binding capacity, affinity for intracellular targets, and tumor cell type [24]. Notably, peripheral benzodiazepine receptors (PBRs), which are highly expressed in malignant tumors such as glioblastoma (GBM), have been associated with the selective accumulation of porphyrin-based compounds. This phenomenon is further supported by the frequent occurrence of microvascular proliferation in tumor tissues, which facilitates enhanced drug uptake [25]. In addition to active uptake, drug efflux mechanisms also play a critical role in determining intracellular photosensitizer concentration. Among these, ATP-binding cassette (ABC) transporters—particularly ABCG2—are known to mediate the efflux of TS from tumor cells, thereby reducing intracellular drug retention [26,27]. The complex interplay between uptake and efflux pathways contributes to the heterogeneous distribution of TS within tumors, which may ultimately influence the effectiveness of photodynamic therapy (PDT). Understanding and modulating these transport mechanisms could provide new avenues for enhancing the intratumoral accumulation of TS and improving therapeutic outcomes. Furthermore, the time interval between TS administration and surgical resection is another important factor that may affect intratumoral drug distribution. Previous studies have shown that TS preferentially accumulates in tumor tissues while gradually clearing from normal tissues over time. Based on this pharmacokinetic profile, we selected a time window of 22–26 h between drug administration and surgery to maximize tumor retention while minimizing systemic exposure. However, we acknowledge that interpatient variability in drug distribution likely exists due to differences in tumor vasculature, permeability, and metabolism. Therefore, future studies incorporating real-time imaging techniques or pharmacokinetic analyses are warranted to further elucidate how variations in the administration-to-surgery interval affect TS distribution and the therapeutic efficacy of PDT. Personalized timing strategies may ultimately enhance treatment consistency and outcomes in clinical practice.

#### 3.2.4. Gene Expression Analysis

RNA-seq analysis revealed significant differences in gene expression associated with intratumoral TS concentration. PDPN, ROMO1, FLOT1, ACTB, HMOX1, PDIA4, and HSPA5 were significantly downregulated in the low-concentration group, whereas DNER was upregulated. qPCR showed discrepancies for RTN4, possibly due to differences in normalization methods, biological variability (tumor heterogeneity, necrosis), technical variations (RNA extraction, reverse transcription efficiency, degradation), and post-transcriptional regulation. Larger sample sizes and standardized methods are required for validation. ACTB and PDPN are highly expressed in invasive tumors [28,29] and were significantly upregulated in high-TS-concentration groups, suggesting a role in cytoskeletal integrity, cellular invasion, and sensitivity to PDT-induced ROS damage [30,31]. ROMO1, associated with lower-grade tumors [32], was downregulated at higher TS concentrations, suggesting favorable oxidative stress conditions and sensitivity to PDT in aggressive lesions. Prior treatments could influence gene expression, potentially obscuring direct correlations with survival [33]. Due to limited sample size (N = 6), these results are preliminary, warranting further validation. TS-induced ROS generation causes ER and lysosomal stress, influencing autophagy and apoptosis pathways [34]. Given the heterogeneous TS distribution and its effect on PDT outcomes, incorporating real-time intraoperative dosimetry (infrared navigation [35], fluorescence, oxygenation monitoring [36] and Monte Carlo modeling [37] may enhance clinical efficacy and treatment personalization.

### 3.3. Summary

As discussed, multiple factors influence PDT efficacy beyond intratumoral drug concentration. Variability in drug uptake, albumin-binding affinity, expression of transporters (e.g., ABCG2), and intracellular (lysosomal) accumulation are significant determinants of PDT effectiveness. However, the outcomes of PDT are not exclusively dictated by these factors. Given the selective accumulation mechanism of TS in tumor tissues, accurate measurement of drug concentration in resected tumors may not reflect the levels present in infiltrative regions. The antitumor effects observed at tumor margins cannot solely be elucidated by drug concentration, as immune responses likely play a critical role in mediating PDT-induced tumor suppression.

Additionally, parameters such as tissue oxygenation, light irradiation parameters, photosensitivity, and cellular immune responses contribute to PDT efficacy. The complex interplay among these biological and environmental variables emphasizes the necessity for a comprehensive approach to optimize PDT protocols. Future research focusing on TS distribution, retention characteristics, and interactions with the tumor microenvironment is essential for improving treatment efficacy. These insights could facilitate the refinement of PDT strategies, supporting personalized treatment methodologies and protocol optimization to enhance clinical outcomes in patients with glioma.

### 3.4. Limitations

First, evaluating the standalone effects of PDT was challenging, as this intraoperative localized therapy is often combined with chemotherapy or other adjunctive treatments for GBM. It remains uncertain whether the observed effects are exclusively attributable to PDT. Second, the tissue samples utilized for drug concentration measurements were not pre-screened for conditions, potentially including necrotic or fibrotic tissues. Although tumor weight was considered in concentration measurements, variability in sample size may have introduced minor errors. Third, this study was limited by sample size, necessitating larger-scale investigations to validate these findings. Fourth, although GBM frequently contains necrotic tissue, the tumor samples were clinically extracted from macroscopically visible non-necrotic areas, thereby minimizing the presence of necrotic components in the specimens. Future studies should isolate PDT from other treatments to assess its standalone effects, pre-screen tissue samples to exclude necrotic or fibrotic tissues, and conduct larger-scale investigations to validate findings.

## 4. Materials and Methods

### 4.1. Patient Selection

The study protocol was approved by the Ethics Committee of Tokyo Women’s Medical University (Approval No. 3540, No. 2022-0180). Informed consent was obtained from all patients included in the study.

Between January 2017 and March 2024, 814 consecutive patients with malignant gliomas underwent surgical resection at the Department of Neurosurgery, Tokyo Women’s Medical University Hospital. Among them, 171 patients who underwent intraoperative TS-PDT received a final clinical diagnosis of grade 4 glioma and provided informed consent were included in the risk factor analysis. Of the 814 consecutive surgical cases, 623 were excluded as combined PDT surgery was not performed. Additionally, from the 191 cases involving combined PDT surgery, 20 were excluded as they did not receive a grade 4 diagnosis on final pathology. No additional exclusion criteria were applied. Grade 4 gliomas were defined as glioblastoma, astrocytoma, IDH-mutant, or diffuse midline glioma with H3 K27M mutation.

Among the 171 patients, consent was obtained for measuring intratumoral TS concentration from only 82 patients (41 newly diagnosed and 41 recurrent cases), with fresh-frozen tumor specimens stored at −80 °C available for analysis. Tumor samples for measuring TS concentration were obtained from the central region of the lesion prior to PDT irradiation. To prevent degradation and potential loss of TS during handling, the samples were immediately frozen at −80 °C following resection with meticulous care. Moreover, to address the heterogeneous intratumoral distribution of TS, the average TS concentration value was used when multiple samples were available, while a single measured value was adopted when only one region could be sampled.

For RNA sequencing (RNA-seq) analysis, specimens from 41 newly diagnosed patients were selected based on specific criteria. The samples had to contain a minimum of 5 mm^3^ of residual tumor tissue suitable for RNA extraction. Three samples with the highest RNA concentrations (0.213 µg/g, 0.555 µg/g, 0.645 µg/g) and three with the lowest concentrations (0.005 µg/g, 0.021 µg/g, 0.029 µg/g) were selected for analysis. In this instance, a sample size of three is justified because it captures extreme variations in TS concentration for exploratory RNA-seq analysis, focusing on significant gene expression differences. Finally, among the patients with measured TS concentrations, three cases with high intratumoral TS concentrations and three with low concentrations that met the RNA extraction criteria were utilized for quantitative polymerase chain reaction (qPCR) to assess gene expression levels (Figure 5).

### 4.2. Data Collection

#### 4.2.1. Risk Factor Analysis

To investigate the relationship between PDT efficacy and various clinical factors, the following data were collected from patient medical records: age, sex, KPS, albumin levels, cholesterol levels, tumor location (temporal lobe, frontal lobe, parietal lobe, and lateralization), genetic profile (IDH, p53, and ATRX mutations), MIB-1 index, history of benzodiazepine use, and smoking history.

Our previous study [8] revealed that PDT significantly improved survival outcomes in patients with newly diagnosed glioblastomas; the PDT group had a median PFS of 19.6 months compared to 9.0 months in the control group (*p* = 0.016) and a median OS of 27.4 months compared to 22.1 months in the control group (*p* = 0.0327). These findings demonstrate that PDT not only extends survival duration but also enhances the progression-free interval in comparison to standard treatment options.

Consequently, in this study, OS data were also obtained. OS was defined as the period from the date of surgery to either the date of death or the date of last confirmed survival.

#### 4.2.2. Relationship Between Intratumoral TS Concentration and Survival

In addition to the aforementioned patient information and survival data, intratumoral TS concentration was measured to assess its correlation with survival outcomes.

### 4.3. PDT

#### 4.3.1. Photosensitizer: TS

TS is a second-generation photosensitizer characterized by faster clearance from the body compared to that of first-generation agents. Its pharmacokinetic parameters include a distribution half-life (t1/2α) of 14.6 ± 2.96 h and an elimination half-life (t1/2β) of 138 ± 21.4 h (mean ± standard deviation). Structurally, it consists of a chlorin backbone derived from plant chlorophyll, with an asparagine residue covalently linked via an amide bond. The presence of asparagine enhances the water solubility of the compound.

Subsequent to TS administration, laser irradiation at a wavelength of 664 nm is applied to the tumor site. The absorbed light excites the TS within tumor cells, leading to the production of singlet oxygen through interactions with molecular oxygen in the tumor tissue. The antitumor effect of TS is primarily linked to the oxidative damage induced by singlet oxygen, which exerts cytotoxic effects through two mechanisms: (i) direct tumor cell damage and (ii) disruption of tumor vasculature.

TS is known to bind to plasma albumin and other high-molecular-weight proteins (65%), high-density lipoprotein (35%), and, to a much lesser extent, low-density lipoprotein (LDL) (1–2%) [16].

#### 4.3.2. Intraoperative TS-PDT Protocol

Intraoperative TS-PDT was conducted utilizing a protocol similar to that previously reported by our group [6,8]. A single intravenous injection of 40 mg/m^2^ TS was administered 22–26 h prior to surgery. Craniotomy was performed under lighting conditions of less than 500 lux. During tumor resection, fluorescence imaging was employed as a control method to assist in identifying tumor margins and guide maximal resection; however, it was not used for tissue sampling. Following maximal tumor resection, verified by contrast enhancement, a semiconductor laser (PD Laser BT; Meiji Seika Pharma Co., Ltd., Tokyo, Japan) with a wavelength of 664 nm was applied to the resection cavity at an irradiation power density of 150 mW/cm^2^ and energy density of 27 J/cm^2^, targeting circular areas with a diameter of 1.5 cm. The laser was aimed at locations considered to have a high recurrence risk, ensuring comprehensive coverage of the resection cavity without overlapping irradiation fields. To minimize light attenuation caused by blood components, such as deoxygenated hemoglobin, the surgical field was thoroughly irrigated with sterile saline before laser irradiation. Additionally, to avoid direct laser exposure to major blood vessels, all visible vessels were carefully covered with aluminum foil. Postoperatively, patients were managed under light-shielding conditions (<500 lux) until skin photosensitivity test results returned negative.

### 4.4. Measurement of TS Concentration

#### 4.4.1. Sample Preparation

The samples were prepared using the QuEChERS method [38]. Distilled water was added to the brain tumor sample to adjust the total weight to 0.5 g, followed by homogenization. An additional 1 mL of distilled water was mixed in, and the mixture was transferred to a centrifuge tube. Subsequently, two stainless steel beads, 1.5 mL of acetonitrile, and 10 μL of fluoranthene (2 ng/mL) as an internal standard were added and mixed thoroughly. QuEChERS extraction salt (0.5 g) was added, followed by vigorous manual shaking and further mixing for 20 s. The mixture was centrifuged at 3500 rpm for 10 min, and the supernatant was carefully transferred to a glass tube using a Pasteur pipette. To the collected supernatant, 100 μL of acetonitrile containing 0.1% trifluoroacetic acid was added and mixed. The sample was then evaporated to dryness under a nitrogen stream at 60 °C. Following the drying process, the sample was cooled to room temperature, resuspended in 200 μL of methanol, and mixed for 5 s. The suspension was transferred to a 1.5 mL microtube and centrifuged at 10,000× *g* for 5 min. Finally, the supernatant was carefully transferred to a measurement vial to avoid sediment uptake.

#### 4.4.2. High-Performance Liquid Chromatography (HPLC) and Settings

Measurements were performed utilizing a Shimadzu liquid chromatography system (LC-2030 3D Plus) equipped with a fluorescence detector (RF-20AXS) and controlled using LabSolutions LC/GC software. A CAPCELL PAK C18 column (150 mm × 4.6 mm I.D.) was used for the analysis. Chromatography was performed under the isocratic mode with an injection volume of 50 μL. The column temperature was maintained at 40 °C, employing water as the rinse solution. The mobile phase consisted of solvent A (2 mmol/L tetrabutylammonium bromide, pH 2.7) and solvent B (methanol) at a flow rate of 1.0 mL/min. The fluorescence detection parameters were set to an excitation wavelength of 405 nm and an emission wavelength of 660 nm, with the cell temperature maintained at 30 °C. Additional settings included a response time of 1.0 s, a sampling rate of 500 ms, a gain factor of ×4, and medium sensitivity. The high-performance liquid chromatography (HPLC) method was validated to ensure the reliability of TS quantification. The limit of detection (LOD) and limit of quantification (LOQ) were calculated based on the standard solution (0.00125 µg/mL) using peak height and noise width in accordance with ASTM guidelines. The noise calculation range was set between 10 and 12 min with a 0.3 min interval, applying a detection limit coefficient of 3.3 and a quantification limit coefficient of 10. Reproducibility was assessed through repeated analyses (*n* = 4) of a standard solution (0.25 µg/mL). The retention times and peak areas demonstrated acceptable repeatability under the analytical conditions. Regarding accuracy, recovery tests were conducted by spiking known concentrations into sample matrices (*n* = 2). The recovery rate was approximately 50%, indicating suboptimal performance. Potential causes for this low recovery included adsorption of TS and internal standards such as fluoranthene during sample preparation, instability of TS, and factors related to sample collection, storage, and handling.

Although no further investigations were conducted to identify the specific causes for reduced recovery, no correction factor was applied. All samples were processed under identical conditions using validated protocols, allowing for reliable relative comparisons across groups despite the recovery limitation.

#### 4.4.3. Calibration Curve

A calibration curve was developed using an internal standard method. Standard solutions of TS at concentrations of 0.00125, 0.00500, 0.01250, 0.05000, and 0.12500 μg/mL were prepared, each containing 1 ng/mL fluoranthene as an internal standard. The peak area ratios of the target compound to the internal standard were computed and plotted against the concentration ratios to generate a calibration curve passing through the origin.

### 4.5. RNA-seq, RNA Extraction, and Library Preparation

Total RNA was extracted from tissue samples using the Maxwell^®^ RSC simplyRNA Tissue Kits (Promega, Madison, WI, USA) in accordance with the manufacturer’s instructions. The quality and quantity of RNA were evaluated using a NanoDrop spectrophotometer (Thermo Fisher Scientific, Waltham, MA, USA) and TapeStation 4150 (Agilent Technologies, Santa Clara, CA, USA). RNA sequencing libraries were prepared using the NEBNext rRNA Depletion Kit v2 (Human/Mouse/Rat) and the NEBNext Ultra II Directional RNA Library Prep Kit following the manufacturer’s protocol. Library quality and concentration were evaluated using LabChip GX Touch HT (PerkinElmer, Shelton, CT, USA) and quantitative PCR (qPCR). Sequencing was performed on a NovaSeq 6000 (Illumina, San Diego, CA, USA) platform, generating 150 bp paired-end reads. Raw sequencing reads underwent adapter trimming and low-quality base removal using Trimmomatic. Clean reads were aligned to the human reference genome (hg38) using HISAT2, and the resulting alignment files were processed with featureCounts (Ver.2.0.6) to quantify gene expression levels. Differential gene expression analysis was conducted with DESeq2, considering genes with an adjusted *p*-value <0.05 as significantly differentially expressed. Pathway enrichment analysis was performed using GOATOOLS (Ver.1.4.12) to identify the biological pathways and processes significantly enriched among the differentially expressed genes.

### 4.6. Real-Time PCR

Total RNA extraction was performed using ISOGEN II (NIPPON GENE Co., Ltd., Tokyo, Japan), and genomic DNA was removed using the Fast Gene RNA Premium Kit (Nippon Genetics Co., Ltd., Tokyo, Japan) to obtain purified total RNA. Quantitative real-time PCR was conducted using a QuantStudio 3 real-time PCR detection system (Thermo Fisher Scientific, Waltham, MA, USA) following the manufacturer’s protocol. PCR conditions included an initial reverse transcription step at 42 °C for 5 min, followed by denaturation at 95 °C for 3 min and amplification comprising 40 cycles of 95 °C for 3 s and 60 °C for 20 s. The relative mRNA expression levels of the target genes were normalized to those of glyceraldehyde-3-phosphate-dehydrogenase (GAPDH) as an internal control. DNA staining during qPCR was performed using the KAPA SYBR Fast One-Step qRT-PCR Kit (Nihon Genetics, Tokyo, Japan). Primers employed for real-time PCR were synthesized by Takara Bio Inc. (Shiga, Japan) and are listed in Table 4.

### 4.7. Statistical Analysis

Group comparisons were conducted using the Mann–Whitney U test for continuous variables and the chi-square test for categorical variables. The evaluation of TS levels incorporated time-to-event analysis to explore their impact on survival outcomes; specifically, we employed Kaplan–Meier curves to stratify survival based on intratumoral TS concentration levels, utilizing a median blood concentration as the cutoff point for both initial and relapsed cases. This is a standard practice in exploratory studies, as information on the effective blood concentration of the drug was not available from previous studies and owing to the insufficient sample size. Log-rank tests were used to compare survival distributions between groups defined by these TS concentration levels. The 95% confidence intervals for the median survival times in the Kaplan–Meier analysis were approximately estimated using the Brookmeyer and Crowley method.

Univariate and multivariate Cox proportional hazards regression analyses were performed to assess risk factors impacting PDT outcomes. Five variables previously suggested to influence prognosis were included in the multivariate analysis: age, KPS, albumin level, history of benzodiazepine use, and smoking history. This comprehensive statistical approach enables us to assess the influence of TS concentration on clinical outcomes while controlling for important confounding variables.

Statistical significance was defined as *p* < 0.05. All reported *p*-values are two-sided. Statistical analyses were performed using JMP Pro 16.0 (SAS Institute, Cary, NC, USA).

## 5. Conclusions

This study established a significant correlation between intratumoral TS concentration and survival outcomes in malignant gliomas, particularly in patients with GBM. Our findings indicate that TS concentration may serve as a prognostic factor for optimizing PDT in grade 4 gliomas. Elevated TS levels were associated with improved survival in recurrent cases, suggesting that factors such as drug uptake, albumin binding, transporter activity (e.g., ABCG2), and immune responses impact PDT efficacy beyond drug concentration alone. These results underscore the necessity for personalized PDT strategies that account for both tumor microenvironmental and molecular factors. Further investigation into TS uptake mechanisms, genetic determinants, and immunomodulatory effects is essential for refining PDT protocols. By integrating these insights, precision-guided PDT approaches can be developed to enhance treatment efficacy and clinical outcomes in patients with glioma, particularly those with recurrent disease.

## Figures and Tables

**Figure 1 pharmaceuticals-18-00583-f001:**
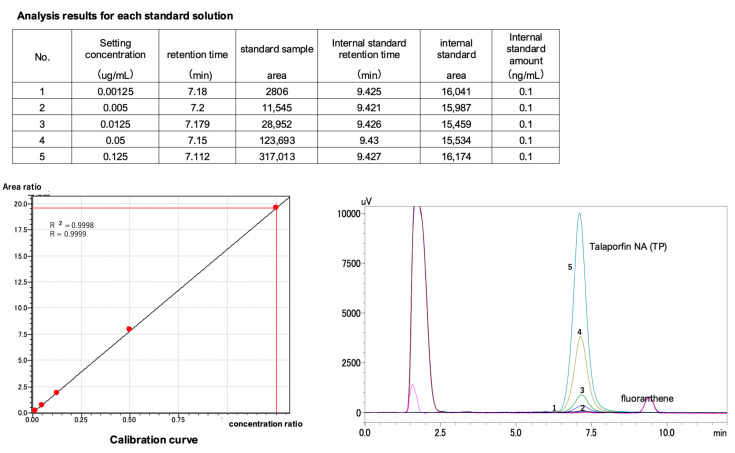
Calibration curve used for concentration calculation. A calibration curve, represented by a straight line passing through the origin, was constructed using the peak area ratio of the standard substance and internal standard substance on the *y* axis and the concentration ratio (sample concentration/internal standard concentration) on the *x* axis. The TS concentration was determined from this curve. The calibration curve has an R^2^ value of 0.999. TS, talaporfin sodium.

**Figure 2 pharmaceuticals-18-00583-f002:**
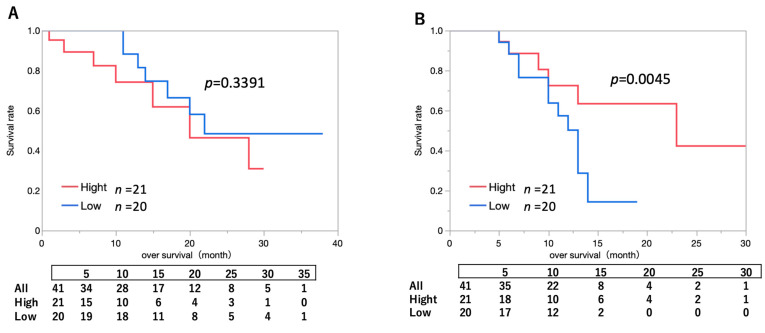
Kaplan–Meier curves for OS for the two groups based on median tissue TS concentrations. (**A**) Newly diagnosed cases: Kaplan–Meier survival curve for OS based on a two-group comparison with a median tissue TS concentration of 0.16333 in patients with newly diagnosed disease. The median survival time was 20.0 months (95% CI: 18.3–21.7) in the High group and 22.0 months (95% CI: 21.4–22.6) in the Low group. The 95% confidence intervals were estimated using the Brookmeyer and Crowley method. (**B**) Recurrence cases: Kaplan–Meier survival curve of OS based on a two-group comparison of the median tissue TS concentration of 0.00883 in patients with recurrence. Patients with median tissue TS concentrations of ≥0.0883 exhibited significantly longer OS than did those with median tissue TS concentrations of <0.0883. The median survival time was 23.0 months (95% CI: 21.9–24.1) in the High group and 13.0 months (95% CI: 12.8–13.2) in the Low group. The 95% confidence intervals were estimated using the Brookmeyer and Crowley method. OS, overall survival; TS, talaporfin sodium.

**Figure 3 pharmaceuticals-18-00583-f003:**
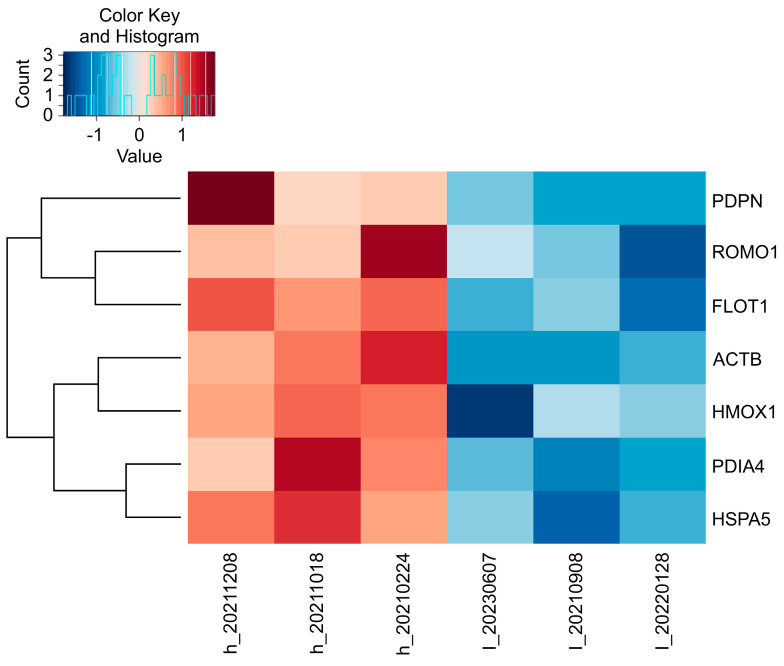
RNA-seq heat map. Differences in gene expression based on high and low tissue drug concentrations were observed for *FLOT1*, *HSPA5*, *ACTB*, *PDIA4*, *HMOX1*, *PDPN*, and *ROMO1*. The expression of these genes was significantly decreased in the low-concentration group (*p* < 0.05).

**Figure 4 pharmaceuticals-18-00583-f004:**
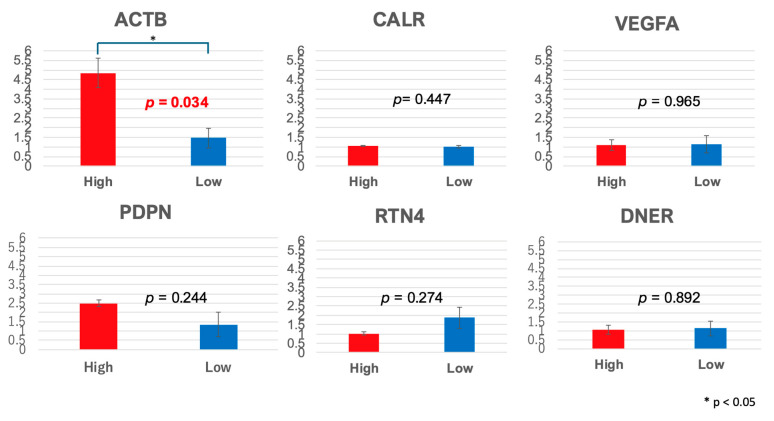
Gene expression levels determined via qPCR. Expression levels of ACTB, CALR, VEGFA, PDPN, and RTN4 were significantly decreased in the group with low intratumoral concentration, as observed in RNA-seq. DNER expression was significantly increased in the group with low intratumoral concentration, as confirmed via qPCR. A significant difference was observed in the expression level of ACTB (*p* = 0.034).

**Figure 5 pharmaceuticals-18-00583-f005:**
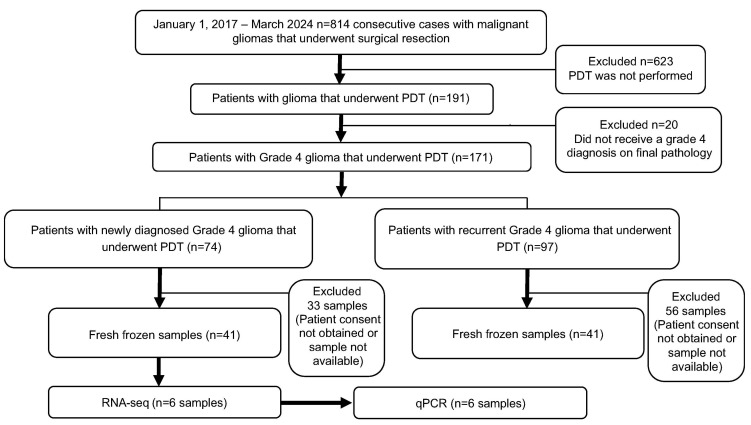
Patient selection flowchart. Between January 2017 and March 2024, 814 consecutive patients underwent surgical resection. Of these, 171 patients clinically diagnosed with grade 4 glioma who underwent intraoperative TS-PDT were included in the risk factor analysis. Among these, tumor specimens from 82 patients (41 initial cases and 41 recurrent cases) were preserved at −80 °C and subjected to intratumoral TS concentration measurements. For RNA-seq analysis, six samples were selected from the 41 initial cases based on the TS concentration. Specifically, the three samples with the highest and three samples with the lowest TS concentrations were selected. The inclusion criterion for RNA-seq was the availability of a tumor sample of at least 5 mm^3^ suitable for RNA extraction. Gene expression analysis was performed on tumor specimens from these six patients categorized by high and low intratumoral TS concentrations. PDT, photodynamic therapy; TS, talaporfin sodium; RNA-seq, RNA sequencing.

**Table 1 pharmaceuticals-18-00583-t001:** Clinical characteristics of patients with grade 4 glioma who underwent PDT between January 2017 and March 2024.

	Newly Diagnosed	Recurrence	*p*-Value
No. of cases	74	97	
Age, in years, mean (SD)	57.5 ± 14.5	47.6 ± 14.0	<0.001
Male/Female sex	42 (56.8%)/32 (43.2%)	70 (72.1%)/27 (27.8%)	0.0360
KPS median (min–max)	70 (40–100)	70 (20–90)	0.6428
Alb median (min–max)	3.2 (1.9–4.3)	3.5 (2.3–4.1)	<0.001
T-Cho median (min–max)	202 (141–279)	201 (105–395)	0.8703
Benzodiazepine	7 (10.0%)/66 (90.0%)	10 (10.3%)/87 (89.7%)	0.8767
Smoking	19 (26.4%)/53 (73.6%)	40 (41.2%)/56 (58.3%)	0.0385
Pathology			
Glioblastoma, *IDH-wildtype*	70 (94.6%)	61 (62.9%)	
Glioblastoma, *IDH-mutant*	3 (4.1%)	23 (23.7%)	<0.001
others	1 (1.3%)	13 (13.4%)	
Localization			
Temporal	31 (41.9%)/43 (58.1%)	27 (27.8%)/70 (72.2%)	0.0548
Frontal	29 (39.2%)/45 (60.8%)	40 (41.2%)/57 (58.8%)	0.7867
Parietal	13 (17.6%)/61 (82.4%)	18 (18.6%)/79 (81.4%)	0.8678
Occipital	5 (6.8%)/69 (93.2%)	4 (4.1%)/93 (95.9%)	0.4475
Right/Left (Bilateral)	43/41 (10)	52/50 (5)	0.5571
Gene			
* IDH (−/+)*	71 (95.9%)/3 (4.1%)	72 (75.8%)/23 (24.2%)	0.0001
* P53 (−/+)*	52 (70.3%)/22 (29.7%)	58 (61.1%)/37 (38.9%)	0.2107
* ATRX (intact/loss/none)*	62(87.3%)/9 (12.7%)/3	67(71.2%)/27 (28.7%)/3	0.0114
MIB-1 (mean ± SD)	24.0 ± 11.3	20.8 ± 13.2	0.1025

Clinical factors, tumor localization, and genetic information of extreme tumors of newly diagnosed and relapsed patients who underwent PDT between January 2017 and March 2024 are shown. Seventy-four patients had initial disease, and ninety-seven had recurrence. MIB-1 and age are expressed as mean ± standard deviation, and KPS, Alb, and T-Cho are expressed as the median (minimum value–maximum value). Tumor localization was classified based on the presence or absence of tumors in each location. For small tumors, both the right and left sides were counted. SD, standard deviation; KPS, Karnofsky Performance Status; Alb, albumin; T-Cho, total cholesterol.

**Table 2 pharmaceuticals-18-00583-t002:** Analysis of risk factors among the two groups of patients who underwent PDT between January 2017 and March 2024: newly diagnosed (*n* = 74) and recurrent (*n* = 97).

	Newly Diagnosed				Recurrence			
	Hazard Ratio (95% CI)	*p*-Value	Hazard Ratio (95% CI)	*p*-Value	Hazard Ratio (95% CI)	*p*-Value	Hazard Ratio (95% CI)	*p*-Value
Age	1.02 (0.99–1.04)	0.26	1.02 (0.99–1.05)	0.27	1.01 (0.99–1.03)	0.22	2.03 (0.81–5.93)	0.50
Male vs. female sex	1.35 (0.62–2.94)	0.44			1.08 (0.62–1.88)	0.45		
KPS	0.98 (0.95–1.00)	0.097	0.96 (0.93–0.99)	0.01	0.09 (0.02–0.33)	0.0003	0.96 (0.95–0.99)	0.0008
Alb	1.51 (0.51–4.48)	0.45	3.68 (1.05–13.76)	0.047	2.22 (0.88–5.84)	0.090	2.13 (0.81–5.93)	0.13
Total cholesterol	1.00 (0.98–1.01)	0.59			1.00 (0.99–1.01)	0.40		
Benzodiazepine	2.36 (0.81–6.88)	0.15	3.06 (0.96–9.76)	0.059	0.77 (0.34–1.72)	0.52	0.77 (0.34–1.74)	0.52
Smoking	1.57 (0.70–3.53)	0.29	1.58 (0.67–3.73)	0.3	1.07 (0.64–1.79)	0.80	1.21 (0.67–2.19)	0.53
Localization								
Temporal vs. not temporal	0.50 (0.22–1.11)	0.079			1.00 (0.56–1.78)	0.99		
Frontal vs. not frontal	1.14 (0.51–2.56)	0.75			0.72 (0.43–1.20)	0.20		
Parietal vs. not parietal	1.32 (0.53–3.31)	0.56			1.36 (0.72–2.53)	0.35		

This table compares risk factors between two groups of patients who underwent PDT: those newly diagnosed (*n* = 74) and those with recurrent disease (*n* = 97). Univariate and multivariate regression analysis results for newly diagnosed cases (*n* = 74) are shown. KPS and Alb values were independent and significant prognostic factors in multivariate analysis. Univariate and multivariate regression analysis results for recurrent cases are shown. When data of patients with recurrence (*n* = 97) were analyzed using the same model as that used for initial recurrence, only a decrease in KPS was identified as a significant risk factor. PDT, photodynamic therapy; KPS, Karnofsky Performance Status; Alb, albumin; CI, confidence interval.

**Table 3 pharmaceuticals-18-00583-t003:** Comparison between the newly diagnosed and recurrent groups based on median tissue TS concentration.

	Newly Diagnosed			Recurrence		
	Mean TS Concentration ≤0.16333	Mean TS Concentration >0.16333		Mean TS Concentration ≤0.0883	Mean TS Concentration >0.0883	
	*n* = 21 51.2 (%)	*n* = 20 48.8 (%)	*p*-Value	*n* = 21 51.2 (%)	*n* = 20 48.8 (%)	*p*-Value
Male vs. female	13 (31.7%)/8 (19.5%)	13 (31.7%)/7 (17.1%)	0.837	13 (31.7%)/8 (19.5%)	14 (34.2%)/6 (14.6%)	0.5843
Age, mean ± SD	60.7 ± 12.2	61.3 ± 11.9	0.884	47.6 ± 14.2	53.6 ± 10.5	0.5843
Left vs. right (excluded bilateral case)	11 (31.4%)/7 (20.0%)	6 (17.1%)/11 (31.4%)	0.1245	10 (26.3%)/9 (23.7%)	6 (15.8%)/13 (34.2%)	0.1869
Pathological diagnosis						
Mean MIB-1 ± SD	28.9 ± 11.9	22.1 ± 11.5	0.0607	21.3 ± 15.4	22.0 ± 15.5	0.8886
ATRX intact vs. loss	20 (51.2%)/0 (0%)	18 (46.2%)/1 (2.6%)	0.226	12 (30.8%)/7 (18.0%)	16 (39.0%)/4 (9.7%)	0.2407
P53 (+) vs. (−)	4 (9.76%)/17 (41.5%)	6 (14.6%)/14 (34.2%)	0.4134	6 (15.0%)/14 (35.0%)	7 (17.0%)/13 (31.7%)	0.7356
IDH (+) vs. (−)	0 (0%)/21 (51.2%)	1 (2.4%)/19 (46.3%)	0.226	5 (12.5%)/15 (37.5%)	4 (9.7%)/16 (39.0%)	0.7047
Part						
Temporal vs. not temporal	7 (17.1%)/14 (34.2%)	9 (22.0%)/11 (26.8%)	0.4435	8 (19.5%)/13 (31.7%)	5 (12.2%)/15 (36.6%)	0.366
Frontal vs. not frontal	11 (26.8%)/10 (24.4%)	8 (19.5%)/12 (29.3%)	0.4261	7 (17.1%)/14 (34.2%)	8 (19.5%)/12 (29.3%)	0.6577
Parietal vs. not parietal	2 (4.9%)/19 (46.3%)	2 (4.9%)/18 (43.9%)	0.959	5 (12.2%)/16 (39.0%)	5 (12.2%)/15 (36.6%)	0.9293

No significant difference was observed in the uptake (concentration) of any factor during the initial recurrence. TS, talaporfin sodium; SD, standard deviation. ATRX, Alpha Thalassemia/Mental Retardation Syndrome X-linked. IDH, Isocitrate Dehydrogenase.

**Table 4 pharmaceuticals-18-00583-t004:** Primers employed for real-time PCR.

Gene		Primer Sequence (5′-3′)
*ACTB*	Forward	TGGCACCCAGCACAATGAA
	Reverse	CTAAGTCATAGTCCGCCTAGAAGCA
*CALR*	Forward	CGAGCCTGCCGTCTACTTCA
	Reverse	CCGTAGAACTTGCCGGAACTG
*VEGFA*	Forward	TCACAGGTACAGGGATGAGGACAC
	Reverse	CAAAGCACAGCAATGTCCTGAAG
*PDPN*	Forward	ATGCCAGGTGCCGAAGATG
	Reverse	TGAAGTTGGCAGATCCTCGATG
*RTN4*	Forward	TGCTGCATCTGAGCCTGTGA
	Reverse	CTCTTGACCAGCCGAAATAGTGTTA
*DNER*	Forward	GGCAACGGTGACACTGCCTA
	Reverse	TGGCATTCCCACAGGCAATA
*GAPDH*	Forward	GCACCGTCAAGGCTGAGAAC
	Reverse	TGGTGAAGACGCCAGTGGA

## Data Availability

The original contributions presented in this study are included in the article.

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
