# Peer review of "Prognostic Factors and Talaporfin Sodium Concentration in Photodynamic Therapy for Recurrent Grade 4 Glioma"

_pharmaceuticals, 2025, doi:10.3390/ph18040583_

Round 1
Reviewer 1 Report
Comments and Suggestions for Authors
Dear Authors,
Thank you for submitting your manuscript, "Prognostic Factors and Talaporfin Sodium Concentration in Photodynamic Therapy for Grade 4 Glioma". I am pleased to inform you that your work is potentially interesting and could be valuable to our readers. However, to ensure it meets the standards of our publication, I would like to offer some constructive feedback that I believe will help you improve the manuscript:
- The introduction should distinctly state the primary hypothesis and research objectives for focus.
- There is no control for dead or scar tissue in TS measurement, potentially impacting data accuracy.
- The ~50% recovery rate lacks corrective actions, which may compromise TS measurement accuracy.
- RNA-seq analysis has only six samples, restricting broad conclusions.
- Small sample sizes in subgroup analyses limit significant result findings.
- Higher TS concentration shows a survival benefit in recurrent patients, but outcomes for newly diagnosed cases are not effective and require clarification.
- qPCR outcomes demonstrate ACTB but differ from RNA-seq for other genes, like RTN4, requiring further discussion.
- Identified genes, such as ACTB, PDPN, and ROMO1, should be better linked to TS uptake and tumor sensitivity to PDT.
- Little discussion exists on real-time monitoring of TS concentration during or before surgery.
- The effect of treatments like steroids and anti-angiogenics on TS uptake and tumor environment is not discussed.
- Some figures, like Kaplan–Meier plots, lack detail, error bars, and high resolution.
- A summary table comparing clinical and molecular findings would improve clarity.
Reference papers:
-Xu, Hua-Zhen, Tong-Fei Li, Yan Ma, Ke Li, Quan Zhang, Yong-Hong Xu, Yu-Cai Zhang, Li Zhao, and Xiao Chen. "Targeted photodynamic therapy of glioblastoma mediated by platelets with photo-controlled release property." Biomaterials 290 (2022): 121833.
-Maity, Surjendu, Tamanna Bhuyan, Christopher Jewell, Satoru Kawakita, Saurabh Sharma, Huu Tuan Nguyen, Alireza Hassani Najafabadi et al. "Recent Developments in Glioblastoma‐On‐A‐Chip for Advanced Drug Screening Applications." Small 21, no. 1 (2025): 2405511.
-Kobayashi T, Nitta M, Shimizu K, Saito T, Tsuzuki S, Fukui A, Koriyama S, Kuwano A, Komori T, Masui K, Maehara T, Kawamata T, Muragaki Y. Therapeutic Options for Recurrent Glioblastoma-Efficacy of Talaporfin Sodium Mediated Photodynamic Therapy. Pharmaceutics. 2022 Feb 2;14(2):353.
I hope you find this feedback helpful. I encourage you to consider these comments as you revise your manuscript. Thank you again for your submission, and I look forward to reviewing your revised version.
Best regards.
Needs improvements.
Author Response
Comments 1: The introduction should distinctly state the primary hypothesis and research objectives for focus.
Response 1: Thank you for pointing this out. We have made the following revisions in the introduction.
before
「This study was conducted with an aim to identify the factors contributing to prolonged survival following PDT and to ascertain whether intratumoral drug concentration correlates with prognosis. By clarifying these factors, we seek to establish a preoperative selection process to identify patients most likely to benefit from TS-PDT.」
after
We hypothesized that certain factors influence the therapeutic efficacy of PDT and intratumoral TS concentration can impact patient prognosis. Accordingly, the objectives of this study were to investigate the factors associated with the efficacy of PDT and to evaluate the correlation between intratumoral TS concentration and clinical outcomes.
“Introduction”
Comments 2: There is no control for dead or scar tissue in TS measurement, potentially impacting data accuracy.
Response 2: Thank you for pointing this out. We have made the following revisions in the limitations section.
although GBM frequently contains necrotic tissue, the tumor samples were clinically extracted from macroscopically visible non-necrotic areas, thereby minimizing the presence of necrotic components in the specimens.
“Limitation”
Comments 3: The ~50% recovery rate lacks corrective actions, which may compromise TS measurement accuracy.
Response 3: We appreciate the reviewer’s insightful comment regarding the ~50% recovery rate and its potential impact on the accuracy of TS quantification.
In our study, we recognized the limitation of incomplete recovery. To ensure the reliability of comparisons, we applied the same sample preparation and analytical procedures to all cases. Although we did not apply a correction factor for recovery loss, all measurements were conducted under the same validated conditions. This consistency allows for meaningful relative comparisons between groups. We have now clarified this point in the Methods section.
Comments 4: RNA-seq analysis has only six samples, restricting broad conclusions.
Response 4: Thank you for pointing this out. We agree with this comment. Therefore, have made the following revisions in 3. Discussion 3.1. Main Results.
RNA-seq analysis revealed a potential enrichment of Gene Ontology (GO) categories related to the endoplasmic reticulum, immune response, and cell membrane endocytosis among the differentially expressed genes. Although the small sample size (n = 6) limits the generalizability of these findings, they offer preliminary insights that may guide future investigations on the relationship between intratumoral TS uptake and immune response in gliomas, as well as the molecular mechanisms underlying the therapeutic effects of PDT.
“3. Discussion 3.1. Main Results”
Comments 5: Small sample sizes in subgroup analyses limit significant result findings.
Response 5: Thank you for pointing this out. We agree with this comment. Therefore, we have changed the following sentences (red text) in 3.2.4. Gene Expression Analysis.
The journal reviewer pointed out 4 corrections in 3.2.4. “Gene Expression Analysis”. We created all different sentences. We have only included a portion of the text because it would be a huge amount of text to include all of it in pharmaceuticals-3556478__2_.docx.
RNA-seq analysis revealed significant differences in gene expression associated with intratumoral TS concentration. PDPN, ROMO1, FLOT1, ACTB, HMOX1, PDIA4, and HSPA5 were significantly downregulated in the low-concentration group, whereas DNER was upregulated. qPCR showed discrepancies for RTN4, possibly due to differences in normalization methods, biological variability (tumor heterogeneity, necrosis), technical variations (RNA extraction, reverse transcription efficiency, degradation), and post-transcriptional regulation. Larger sample sizes and standardized methods are required for validation. ACTB and PDPN are highly expressed in invasive tumors [28, 29] and were significantly upregulated in high TS concentration groups, suggesting a role in cytoskeletal integrity, cellular invasion, and sensitivity to PDT-induced ROS damage [30, 31]. ROMO1, associated with lower-grade tumors [32], was downregulated at higher TS concentrations, suggesting favorable oxidative stress conditions and sensitivity to PDT in aggressive lesions. Prior treatments could influence gene expression, potentially obscuring direct correlations with survival [33]. Due to limited sample size (N = 6), these results are preliminary, warranting further validation. TS-induced ROS generation causes ER and lysosomal stress, influencing autophagy and apoptosis pathways [34]. Given the heterogeneous TS distribution and its effect on PDT outcomes, incorporating real-time intraoperative dosimetry (infrared navigation [35], fluorescence, oxygenation monitoring [36], and Monte Carlo modeling [37] may enhance clinical efficacy and treatment personalization.
“3.2.4. Gene Expression Analysis”
Comments 6: Higher TS concentration shows a survival benefit in recurrent patients, but outcomes for newly diagnosed cases are not effective and require clarification.
Response 6: Thank you for pointing this out. We have added the word “Recurrent” (red highlighted text) in the title. The lack of statistical significance may be attributed to the small sample size. Additionally, in recurrent cases, patients often undergo multiple treatments, which may have altered the tumor microenvironment prior to the measurements. To avoid potential confusion, we have added "recurrent" to the title: Prognostic Factors and Talaporfin Sodium Concentration in Photodynamic Therapy for Recurrent Grade 4 Glioma.
“title”
Comments 7: qPCR outcomes demonstrate ACTB but differ from RNA-seq for other genes, like RTN4, requiring further discussion.
Response 7: Thank you for pointing this out. We agree with this comment. Therefore, we have made the following revisions in the Discussion 3.2.4 (changed sentence of red highlight).
(The journal reviewer pointed out 4 corrections in 3.2.4. “Gene Expression Analysis”. We created all different sentences. We have only included a portion of the text because it would be a huge amount of text to include all of it in pharmaceuticals-3556478__2_.docx.)
RNA-seq analysis revealed significant differences in gene expression associated with intratumoral TS concentration. PDPN, ROMO1, FLOT1, ACTB, HMOX1, PDIA4, and HSPA5 were significantly downregulated in the low-concentration group, whereas DNER was upregulated. qPCR showed discrepancies for RTN4, possibly due to differences in normalization methods, biological variability (tumor heterogeneity, necrosis), technical variations (RNA extraction, reverse transcription efficiency, degradation), and post-transcriptional regulation. Larger sample sizes and standardized methods are required for validation. ACTB and PDPN are highly expressed in invasive tumors [28, 29] and were significantly upregulated in high TS concentration groups, suggesting a role in cytoskeletal integrity, cellular invasion, and sensitivity to PDT-induced ROS damage [30, 31]. ROMO1, associated with lower-grade tumors [32], was downregulated at higher TS concentrations, suggesting favorable oxidative stress conditions and sensitivity to PDT in aggressive lesions. Prior treatments could influence gene expression, potentially obscuring direct correlations with survival [33]. Due to limited sample size (N = 6), these results are preliminary, warranting further validation. TS-induced ROS generation causes ER and lysosomal stress, influencing autophagy and apoptosis pathways [34]. Given the heterogeneous TS distribution and its effect on PDT outcomes, incorporating real-time intraoperative dosimetry (infrared navigation [35], fluorescence, oxygenation monitoring [36], and Monte Carlo modeling [37] may enhance clinical efficacy and treatment personalization.
“3.2.4. Gene Expression Analysis”
Comments 8: Identified genes, such as ACTB, PDPN, and ROMO1, should be better linked to TS uptake and tumor sensitivity to PDT.
Response 8: Thank you for pointing this out. We have made the following revisions (red highlighted text).
(The journal reviewer pointed out 4 corrections in 3.2.4. “Gene Expression Analysis”. We created all different sentences. We have only included a portion of the text because it would be a huge amount of text to include all of it in pharmaceuticals-3556478__2_.docx.)
RNA-seq analysis revealed significant differences in gene expression associated with intratumoral TS concentration. PDPN, ROMO1, FLOT1, ACTB, HMOX1, PDIA4, and HSPA5 were significantly downregulated in the low-concentration group, whereas DNER was upregulated. qPCR showed discrepancies for RTN4, possibly due to differences in normalization methods, biological variability (tumor heterogeneity, necrosis), technical variations (RNA extraction, reverse transcription efficiency, degradation), and post-transcriptional regulation. Larger sample sizes and standardized methods are required for validation. ACTB and PDPN are highly expressed in invasive tumors [28, 29] and were significantly upregulated in high TS concentration groups, suggesting a role in cytoskeletal integrity, cellular invasion, and sensitivity to PDT-induced ROS damage [30, 31]. ROMO1, associated with lower-grade tumors [32], was downregulated at higher TS concentrations, suggesting favorable oxidative stress conditions and sensitivity to PDT in aggressive lesions. Prior treatments could influence gene expression, potentially obscuring direct correlations with survival [33]. Due to limited sample size (N = 6), these results are preliminary, warranting further validation. TS-induced ROS generation causes ER and lysosomal stress, influencing autophagy and apoptosis pathways [34]. Given the heterogeneous TS distribution and its effect on PDT outcomes, incorporating real-time intraoperative dosimetry (infrared navigation [35], fluorescence, oxygenation monitoring [36], and Monte Carlo modeling [37] may enhance clinical efficacy and treatment personalization.
“3.2.4. Gene Expression Analysis”
Comments 9: Little discussion exists on real-time monitoring of TS concentration during or before surgery.
Response 9: Thank you for pointing this out. We have added the following sentences (red text).
Although no clinical systems currently exist to measure intratumoral TS concentration in real-time, our findings indicate the potential clinical value of intraoperative measurement. Given that TS concentration results can be obtained within approximately 1 hour, it may become feasible to estimate prognosis by the end of surgery in recurrent cases.
“3.2.2. Relationship between Intratumoral TS Concentration and Survival”
Comments 10: The effect of treatments like steroids and anti-angiogenics on TS uptake and tumor environment is not discussed.
Response 10: Thank you for pointing this out. In the present cohort, no patients received corticosteroids. Regarding the observed concentration-dependent improvement in survival in recurrent cases, we cannot exclude the potential impact of anti-angiogenic effects from bevacizumab administered prior to recurrence. However, only six patients in this study had received bevacizumab, which was insufficient for meaningful statistical analysis; therefore, this factor was not included in the current discussion. We aim to investigate this further in future studies with a larger sample size.
Comments 11: Some figures, like Kaplan–Meier plots, lack detail, error bars, and high resolution.
Response 11: Thank you for pointing this out. We agree with this comment. Therefore, have made the following revisions. We have improved the resolution of each figure and table. Additionally, we made several modifications to the graphs, including adding significance markers, changing colors, and indicating the number of subjects (n). As for the Kaplan–Meier curve, we initially attempted to display the confidence intervals; however, as it resulted in a visually complex figure, we ultimately decided to omit the confidence intervals from the final version.
“Table and Figure”
Comments 12: A summary table comparing clinical and molecular findings would improve clarity.
Response 12: Thank you for pointing this out. We intend to submit a graphical abstract as a summary table with the manuscript.
“graphical abstract”

Reviewer 2 Report
Comments and Suggestions for Authors
This study is relevant and has significant scientific value, especially in the context of optimizing therapy for malignant brain tumors. Of particular interest are the results of the relationship between the photosensitizer concentration and patient survival.
I have several questions and comments:
- From which area of the tumor were the tumor samples collected (periphery or central area)?
- Were the tumor samples collected before or after PDT?
- Was a control method (e.g. fluorescence imaging) used to identify the area from which the tumor samples were collected? How was the heterogeneity of Talaporfin Sodium distribution within a single tumor taken into account (were several samples collected or only from one area)?
- It is stated that the same laser parameters (power density of 150 mW/cm² and dose of 27 J/cm²) were used for PDT of all patients, without taking into account the intensity of Talaporfin Sodium photobleaching. The authors should discuss the influence of the photosensitizer concentration on the laser propagation in tumor tissues. For example, during PDT, oxyhemoglobin is converted to a deoxygenated form, which, in turn, has an increased contribution of absorption in the 660 nm region. This can reduce the depth of photodynamic action and directly affect the effectiveness of therapy.
- The authors should add to the discussion about the need for intraoperative dosimetry of PDT: for example, Kim, Michele M., et al. "Infrared navigation system for light dosimetry during pleural photodynamic therapy." Physics in Medicine & Biology 65.7 (2020): 075006., Efendiev, Kanamat, et al. "Tumor fluorescence and oxygenation monitoring during photodynamic therapy with chlorin e6 photosensitizer." Photodiagnosis and Photodynamic Therapy 45 (2024): 103969., Finlayson, Louise, et al. "Simulating photodynamic therapy for the treatment of glioblastoma using Monte Carlo radiative transport." Journal of Biomedical Optics 29.2 (2024): 025001-025001.
- Can the authors add their recommendations for optimizing the clinical application of TS-PDT?
- Is there an effect of the time of the start of surgery after the administration of FS on its distribution in tissues. It is stated that the surgery was started 22-26 hours after the start of Talaporfin Sodium administration.
- Were the concentrations of TS measured in the surrounding normal brain tissues?
- Was the possible loss of Talaporfin Sodium during extraction or its instability taken into account when measuring its concentration?
- It is stated that in recurrent tumors, a higher concentration of Talaporfin Sodium was associated with better patient survival (p = 0.0045), but not in primary tumors (p = 0.3391). However, it is not analyzed why this dependence is not observed in primary tumors. Perhaps, these patients have an intact blood-brain barrier (BBB), which limits the penetration of Talaporfin Sodium? Should differences in tumor microenvironment be taken into account: recurrent tumors have increased vascular permeability, which may improve Talaporfin Sodium accumulation. Perhaps the discussion should include data on possible differences in the expression of ABCG2, a transport protein involved in the removal of photosensitizers from cells.
- Could the authors supplement the discussion of the mechanism of action of Talaporfin Sodium. RNA-seq analysis revealed a decrease in the expression of ACTB and PDPN in tumors with low Talaporfin Sodium concentrations, but the possible mechanisms for the association of these genes with PDT are not explained.
- Could differences in liver metabolism in patients affect the level of free Talaporfin Sodium?
Author Response
Comments 1: From which area of the tumor were the tumor samples collected (periphery or central area)?
Response 1: Thank you for pointing this out. The samples in this study were collected from the central region of the tumor. Therefore, have made the following revisions (red text)..
Tumor samples for measuring TS concentration were obtained from the central region of the lesion prior to PDT irradiation.
“5. Materials and Methods.5.1. Patient Selection”
Comments 2: Were the tumor samples collected before or after PDT?
Response 2: Thank you for pointing this out. We have made the following revisions (red text).
Among the 171 patients, consent was obtained for measuring intratumoral TS concentration from only 82 patients (41 newly diagnosed and 41 recurrent cases), with fresh-frozen tumor specimens stored at −80 °C available for analysis. Tumor samples for measuring TS concentration were obtained from the central region of the lesion prior to PDT irradiation.
“5. Materials and Methods.5.1. Patient Selection”
Comments 3: 1、Was a control method (e.g. fluorescence imaging) used to identify the area from which the tumor samples were collected? 2、How was the heterogeneity of Talaporfin Sodium distribution within a single tumor taken into account (were several samples collected or only from one area)?
Response 3: Thank you for pointing this out. Therefore, have made the following revisions in 5.3.2. Intraoperative TS-PDT Protocol and 5.1. Patient Selection (red text)..
- During tumor resection, fluorescence imaging was used as a control method to assist in identifying tumor margins and guide maximal resection; however, it was not used for tissue sampling.
“5.3.2. Intraoperative TS-PDT Protocol”
- Tumor samples for measuring TS concentration were obtained from the central region of the lesion prior to PDT irradiation. To prevent degradation and potential loss of TS during handling, the samples were immediately frozen at −80 °C following resection with meticulous care. Moreover, to address the heterogeneous intratumoral distribution of TS, the average TS concentration value was used when multiple samples were available, while a single measured value was adopted when only one region could be sampled.
“5.1. Patient Selection”
Comments 4: It is stated that the same laser parameters (power density of 150 mW/cm² and dose of 27 J/cm²) were used for PDT of all patients, without taking into account the intensity of Talaporfin Sodium photobleaching. The authors should discuss the influence of the photosensitizer concentration on the laser propagation in tumor tissues. For example, during PDT, oxyhemoglobin is converted to a deoxygenated form, which, in turn, has an increased contribution of absorption in the 660 nm region. This can reduce the depth of photodynamic action and directly affect the effectiveness of therapy.
Response 4: Thank you for pointing this out. It is indeed necessary to consider the influence of photosensitizer concentration on laser propagation within tumor tissues. For example, during PDT, the conversion of oxyhemoglobin to deoxyhemoglobin increases light absorption around 660 nm, which may reduce the penetration depth of photodynamic action and directly affect therapeutic efficacy. It is also conceivable that biological components such as deoxygenated hemoglobin, which absorb light near 664 nm, may be present at the irradiation site. If such substances are abundant, they could interfere with the excitation of Laserphyrin. In light of this, we routinely instruct surgeons to thoroughly irrigate the surgical field with sterile saline to remove blood and other fluids before laser irradiation. In addition, for safety purposes, we advise covering major blood vessels with aluminum foil or similar materials to prevent direct laser exposure. Therefore, we consider it unlikely that interference from blood components near the irradiation site has a significant impact on the clinical efficacy of Laserphyrin. To clarify this point, we have added a corresponding description in the Methods section.
To minimize light attenuation caused by blood components such as deoxygenated hemoglobin, the surgical field was thoroughly irrigated with sterile saline before laser irradiation. Additionally, to avoid direct laser exposure to major blood vessels, all visible vessels were carefully covered with aluminum foil.
“5.3.2. Intraoperative TS-PDT Protocol”
Comments 5: The authors should add to the discussion about the need for intraoperative dosimetry of PDT: for example, Kim, Michele M., et al. "Infrared navigation system for light dosimetry during pleural photodynamic therapy." Physics in Medicine & Biology 65.7 (2020): 075006., Efendiev, Kanamat, et al. "Tumor fluorescence and oxygenation monitoring during photodynamic therapy with chlorin e6 photosensitizer." Photodiagnosis and Photodynamic Therapy 45 (2024): 103969., Finlayson, Louise, et al. "Simulating photodynamic therapy for the treatment of glioblastoma using Monte Carlo radiative transport." Journal of Biomedical Optics 29.2 (2024): 025001-025001.
Response 5: Thank you for pointing this out. We agree with this comment. Therefore, we have made the following revisions in 3.2.4. Gene Expression Analysis.
(The journal reviewer pointed out 4 corrections in 3.2.4. “Gene Expression Analysis”. We created all different sentences. We have only included a portion of the text because it would be a huge amount of text to include all of it in pharmaceuticals-3556478__2_.docx.)
RNA-seq analysis revealed significant differences in gene expression associated with intratumoral TS concentration. PDPN, ROMO1, FLOT1, ACTB, HMOX1, PDIA4, and HSPA5 were significantly downregulated in the low-concentration group, whereas DNER was upregulated. qPCR showed discrepancies for RTN4, possibly due to differences in normalization methods, biological variability (tumor heterogeneity, necrosis), technical variations (RNA extraction, reverse transcription efficiency, degradation), and post-transcriptional regulation. Larger sample sizes and standardized methods are required for validation. ACTB and PDPN are highly expressed in invasive tumors [28, 29] and were significantly upregulated in high TS concentration groups, suggesting a role in cytoskeletal integrity, cellular invasion, and sensitivity to PDT-induced ROS damage [30, 31]. ROMO1, associated with lower-grade tumors [32], was downregulated at higher TS concentrations, suggesting favorable oxidative stress conditions and sensitivity to PDT in aggressive lesions. Prior treatments could influence gene expression, potentially obscuring direct correlations with survival [33]. Due to limited sample size (N = 6), these results are preliminary, warranting further validation. TS-induced ROS generation causes ER and lysosomal stress, influencing autophagy and apoptosis pathways [34]. Given the heterogeneous TS distribution and its effect on PDT outcomes, incorporating real-time intraoperative dosimetry (infrared navigation [35], fluorescence, oxygenation monitoring [36], and Monte Carlo modeling [37] may enhance clinical efficacy and treatment personalization.
“3.2.4. Gene Expression Analysis”
Comments 6: Can the authors add their recommendations for optimizing the clinical application of TS-PDT?
Response 6: Thank you for pointing this out. If intratumoral concentration can be measured intraoperatively and used to predict prognosis, we believe it could contribute to decision-making regarding the extent of tumor resection, particularly in relation to eloquent brain areas.
Comments 7: Is there an effect of the time of the start of surgery after the administration of FS on its distribution in tissues. It is stated that the surgery was started 22-26 hours after the start of Talaporfin Sodium administration.
Response 7: Thank you for pointing this out. We agree with this comment. Therefore, have made the following revisions in 3.2.3. Drug Uptake and Intracellular Dynamics in PDT.
The uptake, intracellular distribution, and retention of porphyrin-based photosensitizers in tumor cells are influenced by a variety of cellular and microenvironmental factors. These include the proliferative state of the cells, binding capacity, affinity for intracellular targets, and tumor cell type [24]. Notably, peripheral benzodiazepine receptors (PBRs), which are highly expressed in malignant tumors such as glioblastoma (GBM), have been associated with the selective accumulation of porphyrin-based compounds. This phenomenon is further supported by the frequent occurrence of microvascular proliferation in tumor tissues, which facilitates enhanced drug uptake [25]. In addition to active uptake, drug efflux mechanisms also play a critical role in determining intracellular photosensitizer concentration. Among these, ATP-binding cassette (ABC) transporters—particularly ABCG2—are known to mediate the efflux of TS from tumor cells, thereby reducing intracellular drug retention [26, 27]. The complex interplay between uptake and efflux pathways contributes to the heterogeneous distribution of TS within tumors, which may ultimately influence the effectiveness of photodynamic therapy (PDT). Understanding and modulating these transport mechanisms could provide new avenues for enhancing the intratumoral accumulation of TS and improving therapeutic outcomes. Furthermore, the time interval between TS administration and surgical resection is another important factor that may affect intratumoral drug distribution. Previous studies have shown that TS preferentially accumulates in tumor tissues while gradually clearing from normal tissues over time. Based on this pharmacokinetic profile, we selected a time window of 22–26 hours between drug administration and surgery to maximize tumor retention while minimizing systemic exposure. However, we acknowledge that interpatient variability in drug distribution likely exists due to differences in tumor vasculature, permeability, and metabolism. Therefore, future studies incorporating real-time imaging techniques or pharmacokinetic analyses are warranted to further elucidate how variations in the administration-to-surgery interval affect TS distribution and the therapeutic efficacy of PDT. Personalized timing strategies may ultimately enhance treatment consistency and outcomes in clinical practice.
“3.2.3. Drug Uptake and Intracellular Dynamics in PDT”
Comments 8: Were the concentrations of TS measured in the surrounding normal brain tissues?
Response 8: Thank you for pointing this out. The TS concentration in the tumor margin was not assessed in this study.
Comments 9: Was the possible loss of Talaporfin Sodium during extraction or its instability taken into account when measuring its concentration?
Response 9: Thank you for pointing this out. Taking into account the potential loss and instability of talaporfin sodium during extraction, tumor samples were promptly frozen at −80 °C following resection to minimize degradation. Therefore, we have made the following revisions in 5. Materials and Methods 5.1. Patient Selection.
To prevent degradation and potential loss of TS during handling, the samples were immediately frozen at −80 °C following resection with meticulous care.
“5. Materials and Methods 5.1. Patient Selection”
Comments 10: It is stated that in recurrent tumors, a higher concentration of Talaporfin Sodium was associated with better patient survival (p = 0.0045), but not in primary tumors (p = 0.3391). However, it is not analyzed why this dependence is not observed in primary tumors. Perhaps, these patients have an intact blood-brain barrier (BBB), which limits the penetration of Talaporfin Sodium? Should differences in tumor microenvironment be taken into account: recurrent tumors have increased vascular permeability, which may improve Talaporfin Sodium accumulation. Perhaps the discussion should include data on possible differences in the expression of ABCG2, a transport protein involved in the removal of photosensitizers from cells.
Response 10: Thank you for pointing this out. We agree with this comment. Therefore, we have made the following revisions in 3.2.2. Relationship between Intratumoral TS Concentration and Survival.
Another factor that may influence intratumoral TS levels is the expression of drug efflux transporters, such as ABCG2, which is known to actively export porphyrin-based photosensitizers from tumor cells, thereby lowering intracellular drug concentrations. Variability in ABCG2 expression between tumors or in response to previous therapies may further contribute to heterogeneous TS distribution and potentially influence PDT efficacy.
“3.2.2. Relationship between Intratumoral TS Concentration and Survival”
Comments 11: Could the authors supplement the discussion of the mechanism of action of Talaporfin Sodium. RNA-seq analysis revealed a decrease in the expression of ACTB and PDPN in tumors with low Talaporfin Sodium concentrations, but the possible mechanisms for the association of these genes with PDT are not explained.
Response 11: Thank you for pointing this out. We agree with this comment. Therefore, we have made the following revisions in 3.2.4. Gene Expression Analysis.
RNA-seq analysis revealed significant differences in gene expression associated with intratumoral TS concentration. PDPN, ROMO1, FLOT1, ACTB, HMOX1, PDIA4, and HSPA5 were significantly downregulated in the low-concentration group, whereas DNER was upregulated. qPCR showed discrepancies for RTN4, possibly due to differences in normalization methods, biological variability (tumor heterogeneity, necrosis), technical variations (RNA extraction, reverse transcription efficiency, degradation), and post-transcriptional regulation. Larger sample sizes and standardized methods are required for validation. ACTB and PDPN are highly expressed in invasive tumors [28, 29] and were significantly upregulated in high TS concentration groups, suggesting a role in cytoskeletal integrity, cellular invasion, and sensitivity to PDT-induced ROS damage [30, 31]. ROMO1, associated with lower-grade tumors [32], was downregulated at higher TS concentrations, suggesting favorable oxidative stress conditions and sensitivity to PDT in aggressive lesions. Prior treatments could influence gene expression, potentially obscuring direct correlations with survival [33]. Due to limited sample size (N = 6), these results are preliminary, warranting further validation. TS-induced ROS generation causes ER and lysosomal stress, influencing autophagy and apoptosis pathways [34]. Given the heterogeneous TS distribution and its effect on PDT outcomes, incorporating real-time intraoperative dosimetry (infrared navigation [35], fluorescence, oxygenation monitoring [36], and Monte Carlo modeling [37] may enhance clinical efficacy and treatment personalization.
“3.2.4. Gene Expression Analysis”
Comments 12: Could differences in liver metabolism in patients affect the level of free Talaporfin Sodium?
Response 12: Thank you for pointing this out. Among the 41 cases investigated in both the newly diagnosed and recurrent groups, 31 cases exhibited liver enzyme levels within the normal range, while 10 cases showed mild elevations corresponding to Grade 1 or lower. Due to the limited sample size, statistical analysis was not conducted to evaluate the relationship between liver function abnormalities and intratumoral drug concentration. We intend to explore this association in future studies with a larger cohort.

Round 2
Reviewer 1 Report
Comments and Suggestions for Authors
I am satisfied with the current form of the manuscript.
Reviewer 2 Report
Comments and Suggestions for Authors
The authors responded to all my comments in a revised version of the article, making this study more informed and structured.